# 6G Enabled Tactile Internet and Cognitive Internet of Healthcare Everything: Towards a Theoretical Framework

**Prafulla Kumar Padhi [1,\*] and Feranando Charrua-Santos [2]**

[1] Department of Industrial Engineering & Management, University of Beira Interior, 6201-001 Covilha, Portugal

[2] Electromechanical Department, University of Beira Interior, 6201-001 Covilha, Portugal; bigares@ubi.pt

\* Correspondence: prafulla.k.padhi@ubi.pt

**Abstract:** Digital era deficiencies traditionally exist in healthcare applications because of the unbalanced distribution of medical resources, especially in rural areas globally. Cognitive data intelligence, which constitute the integration of cognitive computing, massive data analytics, and tiny artificial intelligence, especially tiny machine learning, can be used to palpate a patient's health status, physiologically and psychologically transforming the current healthcare system. To remotely detect patients' emotional state of diagnosing diseases, the integration of 6G enabled Tactile Internet, cognitive data intelligence, and Internet of Healthcare Everything is proposed to form the 6GCIoHE system that aims at achieving global ubiquitous accessibility, extremely low latency, high reliability, and elevated performance in cognitive healthcare in real time to ensure patients receive prompt treatment, especially for the haptic actions. Judiciously, a model-driven methodology is proffered to facilitate the 6GCIoHE system design and development that adopts different refinement levels to incorporate the cognitive healthcare requirements through the interactions of semantic management, process management, cognitive intelligence capabilities, and knowledge sources. Based on the 6GCIoHE system architecture, applications, and challenges, the aim of this study was accomplished by developing a novel theoretical framework to captivate further research within the cognitive healthcare field.

**Keywords:** cognitive data intelligence; cognitive healthcare; tiny machine learning; 6GCIoHE theoretical framework

## 1. Introduction

The shortcomings of the 5G [1] (p. 1) mobile system as a key enabler of the Internet of Everything (IoE) [1] (p. 6) are addressed. To brace the cognitive development of a global ubiquitous smart healthcare system, research activities are being pursued on the next-generation 6G communication system [1] (p. 2). Appendix A provides an acronym list.

The 6G revolution is envisioned to connect everything and control trillions of devices—macro to micro to nano—for the digitization future. Time-sensitive healthcare applications such as haptic (involving touch, sight, and sound) actions and holographic connections displaying three dimensional images assist healthcare professionals using emotion-sensing wearable devices to monitor mental health, heartbeats, oxygen level, glucose, blood pressure, and much more, as shown in Figure 1.

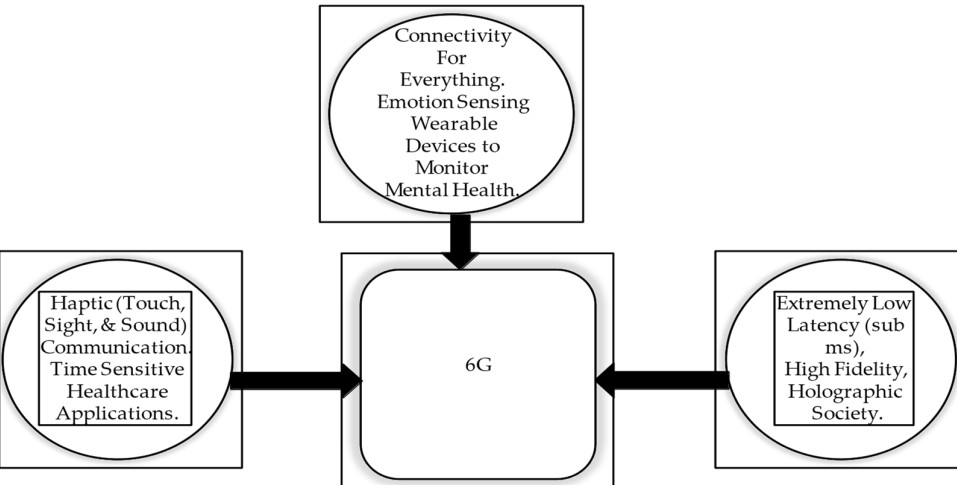

**Figure 1.** Examples of 6G for healthcare applications.

The Tactile Internet [2,3] bestows a new paradigm in people to machine (P2M) interaction featuring extremely low latency and short transit, high availability, great reliability, and an enormous level of security to create systems with real time interactive actions. Currently, based on the 5G advancements and deployment, the stage is being set for the emergence of 6G enabled Tactile Internet [1–3], a new paradigm, to provide the integration of mobile edge-clouds and eXtended Reality [1] (p. 5) (eXR—augmented reality—AR, virtual reality—VR, and mixed reality—MR) for sensory and haptic controls to address the future smart healthcare systems with less than one millisecond reaction time, offering physical tactile experiences remotely.

Cognitive technology (CT) [4] is a discipline of computer science that mimics human brain functions through various means such as cognitive computing (CC) [4] and tiny machine learning (tiny ML) [5]. CC is a propitious tool for analyzing massive data for better holistic ideas to help healthcare businesses accomplish faster and smarter decisions. Tiny AI is an emerging avant-garde discipline to revolutionize the healthcare industry that intersects between tiny ML and embedded IoE devices and that requires intimate knowledge of hardware, software, algorithms, and applications. The integration of CC, massive data analytics (MDA) [6], and tiny ML constitutes cognitive data intelligence (CDI) [4–6] to offer better healthcare solutions by providing relevant information to assist stakeholders in decision-making.

The Internet of things (IoT) [1] is an online, dynamic, rapidly changing world that empowers industries such as healthcare. As IoT contains billions of devices, the spectrum usage efficiency faces a daunting challenge in harboring that many devices within the sparse spectrum. The IoE is a superset of IoT and is replaced by the IoE era, where ubiquitous connectivity is essential. The IoE connectivity is a discipline that provides various solutions for massive data to be exchanged to the edge of the cloud infrastructure. With the emergence of the IoE era, industrial control systems (ICSs) [1] are switching into the Industrial Internet of Everything (IIoE) [1] for various industries, especially for healthcare. The Internet of Healthcare Everything (IoHE) [1] is a sub-classification of IoE/IIoE which explains uniquely identifiable devices connected to the 6G enabled Tactile Internet that are able to communicate with each other for smart healthcare applications, enabling localization and real-time information about healthcare stakeholder assets. In this study, IoT/IoE and IoHE terms are used interchangeably.

To satisfy ever-expanding bandwidth demand, the CT-based IoHE, namely cognitive IoHE (CIoHE) [1,4], is an assuring capability and approach for healthcare professionals to collect the real time physiological data of the patients. Thus, the integration of 6G enabled

Tactile Internet, CDI, and IoHE formalizes the 6GCIoHE system [1–6], which plays a vital role in cognitive healthcare applications.

The primary purpose of this study was to develop a novel theoretical framework based on the 6GCIoHE system architecture, applications, and challenges that lie at the heart of the cognitive healthcare domain.

### 1.1. Background and Problem Formulation

At present, the healthcare sector is facing numerous challenges. The deficiencies of the 5G mobile system as an enabler of IoE have inspired global research activities to focus on the 6G wireless system. The advancement of the 6G Tactile Internet, CDI, and IoHE can play a pivotal role in providing interoperability for healthcare applications to alleviate the challenges of the present healthcare system. Hence, we propose a holistic approach to form the 6GCIoHE system for improving the healthcare domain. Academic researchers assign an esteemed value to theory development. Although there is an increasing interest in comprehending theory, the process of theory advancement is rarely addressed with contributions from other subjects or topics to harmonize them meaningfully. To date, in the context of the 6GCIoHE system, there is neither a central corpus of a well-accepted theory nor a theoretical framework in the literature. Because of the theory development void in the literature related to the 6GCIoHE system for healthcare applications, this study offers comprehensive new knowledge to develop a novel theoretical framework with a significant contribution of value.

Although the issues of privacy, security, and trust were extensively discussed in the literature, a gap exists in establishing the CIoHE global standards. The CIoHE ought to be part of a global healthcare ecosystem to evaluate security, privacy, and trust considerations and not splinter into inconsistent sets of rules or standards. One must engage with healthcare stakeholders to support the development of global standards to foster innovation and promote security and privacy solutions.

### 1.2. Significant Contributions

The contributions and the significance of this study are as follows: (i) the study discusses the foundations of the CIoHE framework, which helps improve services and experiences, reduce costs, and promote next-gen healthcare facilities and automates the workflow of patient care; (ii) for the first time, the enabling technologies that constitute the 6GCIoHE system architecture demonstrate machine to machine communication, haptic actions, interoperability, data movement, and information exchange that allows the healthcare industry to deliver efficiency; (iii) we describe the opportunities and the challenges that emerge in executing the 6GCIoHE system, including the CDI scalability requirement and the concerns of data security, privacy, as well as risk perceptions. Thus, the primary contribution of this study was achieved by developing the novel 6GCIoHE theoretical framework and bestowing new knowledge in the literature.

*1.3. Organization summary.*

The structure of the paper comprises of the following sections and the related topics:

| Sections | Topics |
|---|---|
| Literature review | 6G enabled Tactile Internet, CDI, and IoHE. |
| Theoretical foundation | Research boundary, related frameworks, and relevant theories. |
| Methodology | Model-driven methodology mapping 6GCIoHE system. |
| 6GCIoHE system paradigm | Enabling technologies. |
| 6GCIoHE architecture | A 5-layer modular architecture is proposed. |
| Applications and challenges | 6GCIoHE system for smart healthcare. |
| Theoretical framework | Building blocks of the 6GCIoHE system theoretical framework. |
| Consolidated lessons learned | 6GCIoHE system for next-gen healthcare. |
| Recommendations | For academia and practitioners. |
| Conclusion | Theory's prodigy not covered by prior theories. |

## 2. Literature Review

The literature review and the theoretical framework are intrinsically linked for logically understanding and developing the disparate yet interconnected essential parts of the literature review.

A thorough review of relevant literature in any study undertaking bestows a sound foundation for actuating new knowledge and creates theory development as well as identifies where the research gap exists. The goal was to enhance the comprehension of the 6GCIoHE system technologies that support software, hardware development, testing, and evidence of its use in the healthcare sector. The investigation of topics, research directions, and insights, as shown in Tables 1 and 2 included the following: (i) the theoretical foundation of technologies related to 6G, Tactile Internet, CDI, IoHE; (ii) the 6G Tactile Internet and the IoE systems ascendancy leveraged by software defined networking (SDN) and network function virtualization (NFV) technologies [7], strengthening their comprehensive security, resilience, and flexibility, thus providing substantial added value; (iii) the Tactile Internet for smart healthcare in the 6G domain for ultra-reliability and fast response time; (iv) the 6G enabled Tactile Internet as the next revolution for the haptic communication system; (v) the challenges and the standards for the Tactile Internet in the 6G domain; (vi) the future of population health management (PHM) [8] (p. 10) trussed to cognitive computing, which converts unstructured data into structured data by utilizing massively parallel processing and tiny ML; and (vii) cognitive applications where clinicians can collaborate with cognitive computing systems to improve healthcare.

The following steps were taken to investigate the relevant literature: (i) usage of the key terms 6G, CC, MDA, tiny ML, and IoHE; (ii) review of Google, Firefox, and leading journals such as IEEE, MDPI, Science Direct (Elsevier), and Emerald Insight was used related to 6G, Tactile Internet, CDI, IoHE, and healthcare applications as well as challenges; (iii) search of academic research databases related to 5G, 6G, Tactile Internet, CC, tiny ML, CDI, IoHE, and associated theories; (iv) industry white papers were used related to the research paper themes; and (v) security, risk perception, and privacy matters concerning trust in the IoHE were investigated.

**Table 1.** Summary of the 6G/Tactile Internet/CDI/IoHE literature review.

| References | Research Directions/Insights |
|---|---|
| Alsharif et al. [9] | 6G Networks Research Activities |
| Dang et al. [10] | 6G Human Centric Perspective |
| Gui et al. [11] | 6G New Horizon and Security |
| Chowdhury et al. [12] | 6G Applications and Research Directions |
| Janbi et al. [13] | 6G/Smarter IoE/AI |
| Zhang et al. [14] | 6G/Super IoT |
| Ateya et al. [15] | Towards Tactile Internet |
| Simsek et al. [16] | 5G Enabled Tactile Internet |
| Antonakoglou et al. [17] | Haptic actions over the 5G Tactile Internet |
| Dai et al. [18] | IoMT to Combat COVID-19 |
| Atlam et al. [19] | IoT and AI |
| Cisotto et al. [20] | Healthcare Services over Cellular Systems |
| CISCO [21] | IoE Economy |
| Siemens [22] | Industrial IoT 2050 |

**Table 2.** CC, AI, and ML theoretical framework (TF) review.

| References | Research Directions/Insights |
|---|---|
| Wan et al. [23] | CC and wireless communications for healthcare |
| Bini S.A. [24] | AI, ML, Deep Learning, and CC |
| Norden et al. [25] | Promise of CC: Ushering in a New Era |
| Khalil et al. [26] | CC and the Future Healthcare: IBM Watson |
| Polson et al. [27] | Cognitive theory-based of user interfaces |
| Nord et al. [28] | IoT/TF/Systematic Review |
| Falkenberg et al. [29] | Information System (IS) Theory/Limitation |
| Aquilani et al. [30] | Value Creation TF |

## 3. Theoretical Foundation

### 3.1. 6GCIoHE System Boundaries and Research Approach

Reflecting on the history of technology accepting theories, scholars have understood the evolution and the development of such theories. Although there has been an increase in comprehending theory, the process of theory advancement is addressed rarely with contributions from other disciplines to harmonize them meaningfully. The reason behind accepting or rejecting any new technology investigation remains the most censorious area in the information communication technology (ICT) discipline. Technology acceptance theories aim to express the concept of the user's knowledge of how one would use such technology. Most technology theories were introduced to measure the degree of satisfaction and acceptance toward any technology of ICT system. Therefore, for any novel technology execution, there would be many variables that influence the execution of a specific technology.

Theories of technology describe the factors that shape technological innovation and the impact of technology on society. From the perspective of ICT, there are four theories that have frequently been utilized in the advent of behaviorism, cognitivism, constructivism, and connectivism as learning theories for the digital age. The aims of ICT theory are threefold: (i) to provide a comprehensive and compelling overview of cutting-edge ICT; (ii) to identify and discuss the fundamental principles underlying technologies such as CDI, 6G Tactile Internet, and IoHE; and (iii) to investigate the reciprocal relationship between these technologies and societies [31].

It is critical to lay higher foundations for theory building from a greater cumulative and coherent research base. Consequently, this research considered the underlying

challenges in research of ICT and reflected especially on the 6GCIoHE system to reflect the principles of the disciplined inquiry consisting of cognitive healthcare applications.

### 3.2. 6G Enabled Tactile Internet Framework for Healthcare

The theoretical foundation of wireless communication is based on fundamental principles from communication and information theory, signal processing, detection, and estimation. The 6G framework is an emerging network technology, and distinct devices characteristically evolve to the subsequent paradigm value-added services and are achieved through the development of a distributed service 6G enabled Tactile Internet framework, as shown in Figure 2, that encourages tactile apps for the healthcare industry.

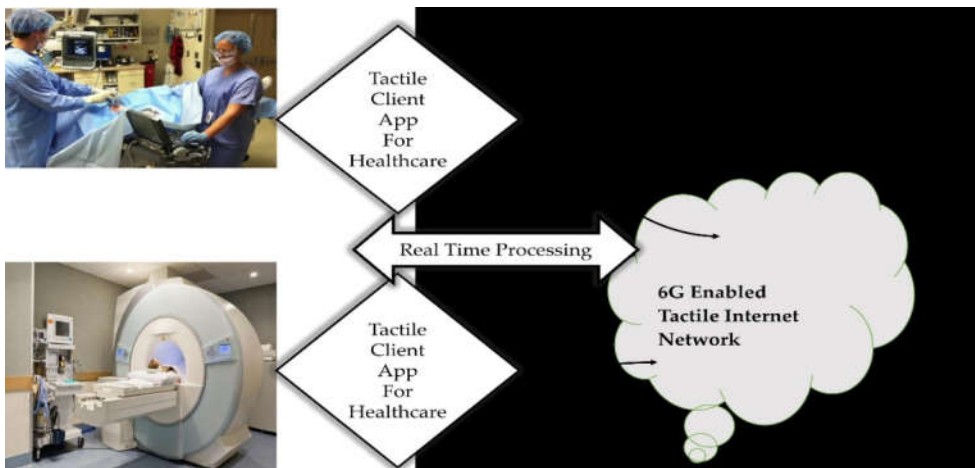

**Figure 2.** 6G enabled Tactile Internet framework for healthcare apps.

### 3.3. The IoHE Conceptual Framework in the Healthcare

The significant and most widely used theories for technology acceptance that have been used to study IoT adoption by a great deal of peer-reviewed research are explained using sociology and psychology models associated with six independent variables, as shown in Figure 3, and are as follows: (i) performance expectancy; (ii) effort expectancy; (iii) facilitating condition; (iv) social influence; (v) perceived credibility; and (vi) attitude.

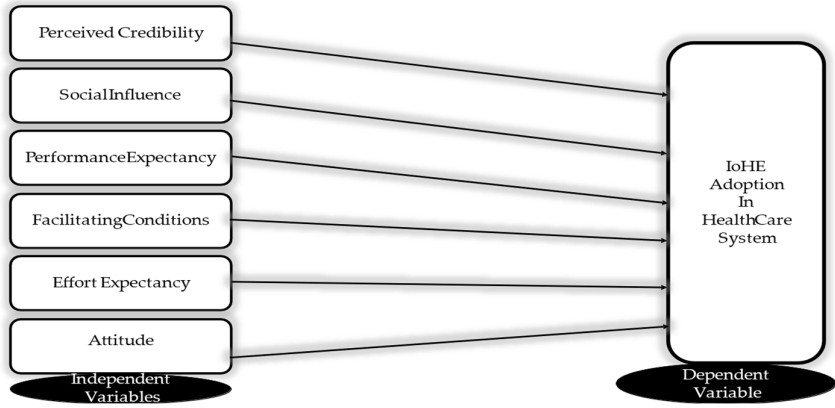

**Figure 3.** Conceptual framework for the adoption of IoHE in healthcare.

Considering the above, the IoHE conceptual framework was found to be most suitable for establishing relationships and outcomes between independent and dependent variables, which in turn affect the user behavior of the healthcare services.

*3.4. The IoHE Pillars in Healthcare*

The IoHE pillars comprise data, things (sensors), people, and processes, as shown in Figure 4, that are intelligently connected with billions of sensors to distinguish calibration and appraise their conditions.

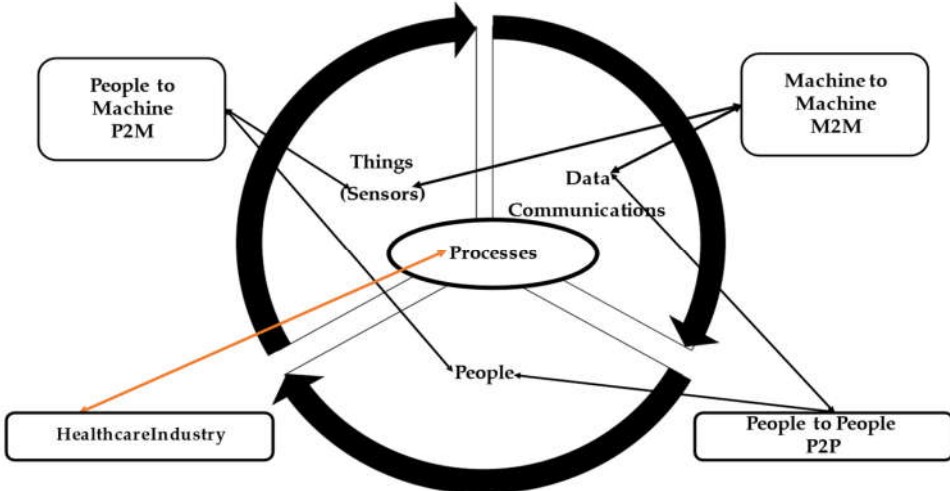

**Figure 4.** IoHE Pillars in healthcare.

The "Process" is key among "people, data, and things", which interact with each other to deliver value for bestowing immersive patient experience.

### 3.4.1. Sensors

Medical sensors are the eyes and the ears of the IoHE network. Sensors convert physical signals into electrical signals. As wireless body sensors (WBS) including medical super sensors (MSS) sense data from the patient, the data need to be transmitted to the processing node via the communication network. The functioning IoHE technology depends upon precise measurement by medical sensors. Therefore, it is vitally significant to select appropriate sensors suitable for a healthcare task.

### 3.4.2. Communication

M2M communication is the key technology behind the IoHE. Values that are sensed by the medical sensors need to be sent for the computing. With the help of 6G Tactile Internet communication, it is easy to establish communication among the various networks using standard IP protocols.

### 3.4.3. Cognitive Computing (CC)

CC is taking center stage for healthcare, helping to address issues around privacy, and refers to the process of analyzing a problem the way the human brain does. Advances in sensor technology and tiny ML and IoHE devices are integrated to mimic the human brain in solving problems. CC in an IoHE system enables the analysis of hidden patterns that are included in a massive amount of data and improves the ability of a sensor to process healthcare data. The functioning of humans and machines where computers and the human brain truly overlap can be used to improve human decision-making in healthcare applications, bestowing significant value by enhancing the quality of patient care. Single-chip CC is called a microcontroller that functions at the speed of tera- or gigahertz, with low power consumption necessary for the IoHE applications.

### 3.4.4. Data Analysis

Based on the billions of sensors utilized, the sensors generate massive data. With the mining of a massive amount of data availability and technologies, the world is witnessing the next phase of healthcare. Cloud computing is needed to store the data, but the difficult part is doing an analysis of massive data and identifying useful data. This process is strenuous and developing smart algorithms to gain useful information in real time is a challenging task. The application of MDA and tiny ML can reduce the complexities and the uncertainties associated with the healthcare industry.

### 3.5. CDI Framework

CDI, which constitutes the integration of CC, tiny ML, and MDA, can be used to palpate a patient's health status, physiologically and psychologically transforming the current healthcare system.

### 3.5.1. CC, as a Sub-Field of AI

CC applications in healthcare produce superior, best-practice, decision-relevant information for everyone, and computing is analogous to human cognition, rationale, and judgment and has the capacity to deal with symbolic and conceptual information to offer a path to more individualized healthcare decisions.

### 3.5.2. Tiny ML

Tiny ML is an emerging discipline to revolutionize the healthcare industry that intersects between ML and embedded IoHE devices. The ability to distribute AI resources to memory-constrained devices could have significant advantages for the healthcare sector. Figure 5 shows the AI systems with tiny ML achieving efficiency improvement in computation, engineering manpower, and data efficiency.

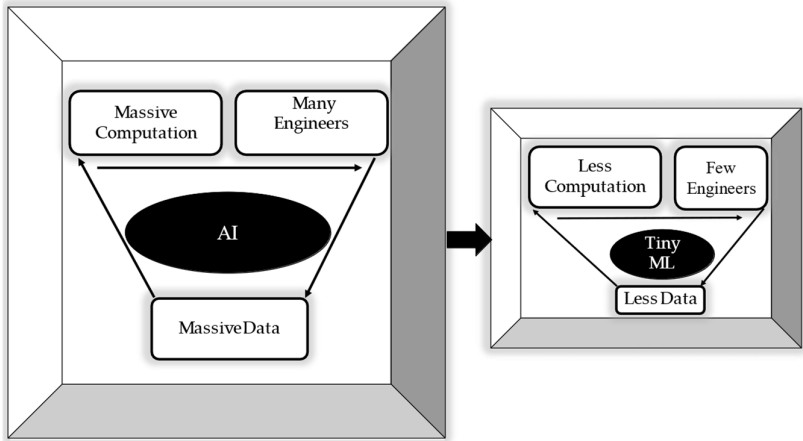

**Figure 5.** From AI to tiny ML with significant benefits for healthcare systems.

This study proposes a novel tiny ML healthcare framework that includes the following levels:

Level 1: Integrate tiny ML techniques into standard analytic tools for one-click access.

Level 2: Select or the tailor predefined predictive models or build a customized model.

Level 3: Optimize predictive models to maximize insight and drive better decision-making.

Level 4: Compare performance over time or against peers to refine improvement interventions.

Level 5: Learn from data to drive future improvements.

### 3.5.3. MDA

MDA utilizes huge multiple sets of real time data through tiny ML-based algorithms in this healthcare system to provide a better service for the welfare of society, revolutionizing the healthcare industry.

Tiny ML- based MDA rapidly translates the core areas of a patient's health conditions with comprehensive and accurate reports. There are enormous sources available for MDA to analyze for future endeavors such as electronic health records (EHR), social media platforms, mobile apps, pharmaceutical research, payer records, campaigns, and smart medical devices. Due to the availability of high-quality real time data, the results generated from MDA are more accurate and effective. While medical professionals cannot make sense of billions of data points across millions of patients, tiny ML can improve healthcare with more types of data and more cases to process. Cognitive healthcare has dual benefits: (i) improving patient health and (ii) reducing the costs of sub-optimal treatment plans.

Based on subsections 3.5.1, 3.5.2, and 3.5.3 mentioned above, Figure 6 shows the basic elements of the CDI framework that include understanding, verifying, planning, evaluating, attention, and perception to serve as objectives in a specific cognitive task.

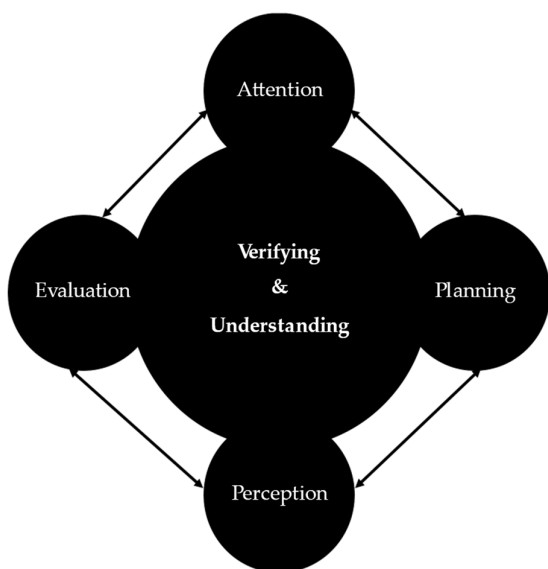

**Figure 6.** Basic CDI framework.

### 3.6. Relevant Theories Supporting the Theoretical Foundation

Since a theoretical framework is defined as the set of existing theories, models, concepts, and relevant definitions that are used in a specific field of study, we structured the theoretical foundation based on the following relevant theories as the proof to support and hold the 6GCIoHE ecosystem for healthcare.

### 3.6.1. Theory of Mind (ToM)

ToM is a psychological construct that displays the capacity for characteristic thoughts, feelings, and ideals toward others and the ability to apprehend different humans' intellectual states. ToM and intelligence are stimulated through domain-precise tasks, and overall performance is associated with numerous cognitive capacities [32].

### 3.6.2. Social Cognitive Theory (SCT)

SCT indicates a triadic relationship among the individual, the individual's private behaviors, and outside factors. It is used to apprehend behaviors amongst humans guided through functions and objectives which can be encouraged via their private ideals of self-

efficacy and through purpose expectancies from their behaviors within a selected environment. Research affirms the uses and gratifications paradigm and increases it to a theory of the Internet with attendance grounded in SCT [33].

### 3.6.3. Theories of Inference and the Mathematics of Probability (TIMP)

TIMP applications are for data statistics evaluation. None of the classical theories of statistics come near discovering new data in an actual scientific problem. Most formal theories, including Bayesian, decision-theoretic, Neyman–Pearson, and others, work with prespecified possibility models. In practice, hypotheses frequently emerge after the data have been examined [34].

### 3.6.4. The Interface Theory (TIT)

TIT investigates an architecture for IoT applications where so-called "accessors" offer an actor-oriented proxy for devices ("things") and services, and the composition of those interfaces permits the combination of a timed actor model and the pattern, enabling careful assessment of design selections for the 6GCIoHE applications, where "everything" and services interact with the physical world [35].

### 3.6.5. The Communication Theory (TCT)

TCT studies the process of information, interdisciplinary disciplines of interpersonal communications, psychological paradigm, and philosophical and social dimensions [36].

### 3.6.6. The Systems Theory (TST)

TST is implemented in the 6GCIoHE systems for evaluation applications. One of the essential mechanisms of systems analysis is systems thinking, enabling the contouring of systems from a broad perspective as opposed to precise activities within the system [37].

### 3.6.7. The Theory of Transparency (TOT)

The TOT wherein perpetual experience is frequently stated to be transparent and privy to the properties of the objects around an environment, and the fundamentals of transparency require actions and decisions. Since a theoretical framework is defined as the set of existing theories, models, concepts, and relevant definitions that are used in a specific field of study, we structured the theoretical foundation based on the following relevant theories as the proof to support and hold the 6GCIoHE ecosystem for healthcare [38].

### 3.6.8. Ethical Theory (ET)

The ET offers moral concerns inherent to all the stages of the healthcare sector and drives the ethics of care based primarily on ethical interpersonal relationships and care as a virtue [39].

## 4. Methodology

As the healthcare industry is under pressure for return on investment and deficiencies in digital technology for healthcare applications, there is an extensive effort from the research community for novel and profitable automation process management as well as semantic management of healthcare platforms. A new era of the IoHE so-called "Cognitive IoHE" was introduced that aims at integrating cognitive technologies into the IoHE-based systems to ensure smart management through enabling cooperation and interaction between the IoHE and humans. Autonomic CC sheds light on unprecedented opportunities for developing smart CIoHE systems and a strong focus on managing complex systems through automating tasks based on specific patterns (monitoring, analysis, plan, execution, and knowledge).

Considering the above, we introduced a collaborative model-driven methodology [40] to map the 6GCIoHE system functions which monitor and perform analysis, planning, and execution of the management process. Within this methodology, a set of cognitive design patterns was suggested to drive the 6GCIoHE system architect to provide flexible, smart IoHE-based applications. These patterns deal with functional and non-functional requirements, ensuring the management of CDI capabilities and scalability issues within a smart healthcare system. The patterns need smart manageability, interoperability, and scalability of the 6GCIoHE system. These patterns are represented to design the structure, to draw the behavior, and to delineate how the management processes should be coordinated to meet the system's functional requirements based on the knowledge pattern. Once the management processes are identified and modeled, the next level is semantic integration management, where mainly information about the system and its environment as well as procedural knowledge (know-how) for decision-making is formalized to be automatically reused by the management processes. However, the ability to manage IoHE data variety, velocity, volume, and system performance in terms of response time and scalability management is a paramount concern. Data quality is crucial, therefore, five traits within data quality—accuracy, completeness, reliability, relevance, and timeliness—are imperative. Figure 7 portrays an overview of the proposed methodology that includes two phases: (i) identification and (ii) formalization.

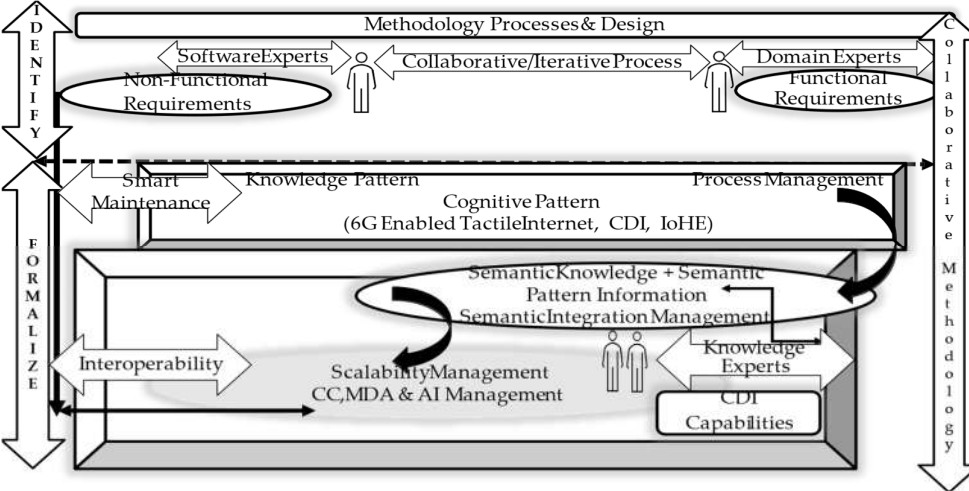

**Figure 7.** A model driven methodology for the design of 6GCIoHE system.

### 4.1. Identification

Identification is to retrieve the system functions and to identify the non-functional requirements. It is an iterative process, where the functional requirements are incrementally refined and represented using a use case describing the functions of the system without specifying any implementation details.

### 4.2. Formalization

Formalization is to focus on formalizing and structuring the identified requirements into concrete models describing the system processes' interactions. Within the formalization phase, sub-levels are introduced incrementally to deal with challenges related to the design of the 6GCIoHE system such as the coordination of management processes and the semantic integration of CDI and IoHE.

## 5. 6GCIoHE System Paradigm

### 5.1. The Next Leap Evolution

As shown in Figure 8, the Tactile Internet adds a new dimension to P2M interaction by tactile and haptic sensations, enabling an unforeseeable plurality of new applications, products, and services. Ubiquitous availability, better reliability, and high security with extremely low latency will define the character of the emerging 6G enabled Tactile Internet, introducing numerous new opportunities.

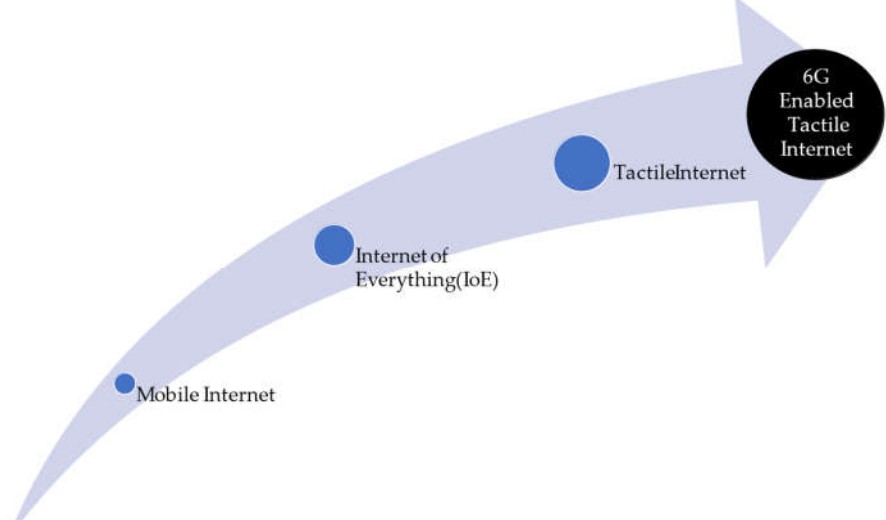

**Figure 8.** Leap evolution of 6G enabled Tactile Internet.

The 6G vision connectivity, as shown in Figure 9, is to brace the development of a global, ubiquitous intelligent mobile society (GUIMS) with the following achievements: (i) solve the limitations of 5G including system coverage and IoE; (ii) achieve up to a hundred times higher data rate, greater system capacity, better spectrum efficiency, and lower latency than 5G to serve the interconnection of everything; (iii) introduce ubiquitous, intelligent, and integrated network with holographic, broader, and deeper coverage, including terrestrial communication, satellite communication, and device-to-device communications;

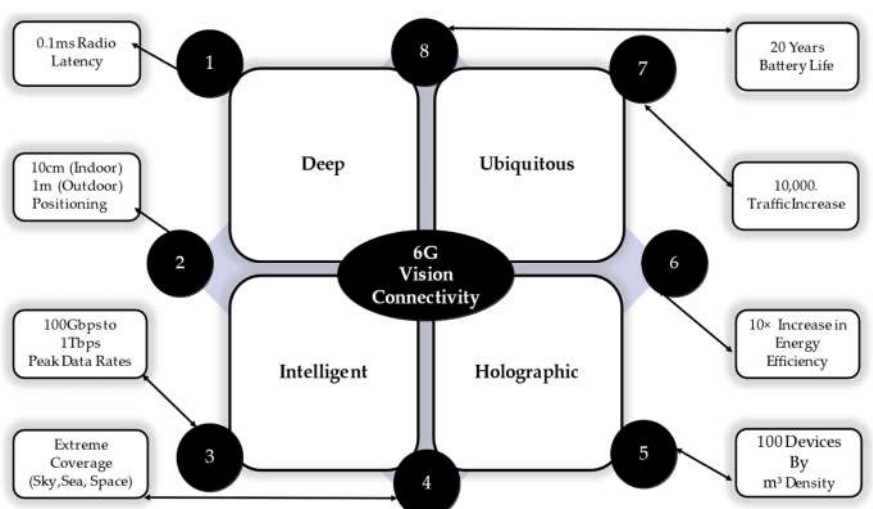

**Figure 9.** 6G vision connectivity.

(iv) serve airspace, land, and sea, realizing a global mobile broadband communication system; (v) work on higher frequency to achieve wider bandwidth, such as better mmWave, terahertz, visible light, etc.; (vi) produce a personalized intelligent network combined with artificial intelligence (AI), especially tiny ML, to offer virtualized personal mobile communication with an endogenous security scheme or function security to offer the capability of self-awareness, real time dynamic analysis, and confidence evaluation, realizing cyberspace security; (vii) merge computation, navigation, and sensing with communications; (viii) adopt a more open architecture with software defined network (SDN), virtualized network functions (VNF) and radio access network (RAN) to realize self-intelligent development and rapid dynamic deployment of network functions; (ix) generate massive data through the IoE combining with novel technologies such as cloud computing, edge computing, tiny machine language (ML), blockchain, etc., realizing group collective intelligence (swarm intelligence); (x) act as a "global wireless power grid"; and (xi) develop a better massive MIMO.

In the emerging 6G and beyond of wireless communication, an increasing number of ultra-scale intelligent factors will result from interference exploitation. To manage this exploitation, challenges for detection algorithms in uplink MIMO systems exist, especially for higher-order quadrature amplitude modulation (QAM) signals.

### 5.2. The 6G Enabled Tactile Internet End-to-End Architecture

As shown in Figure 10, the architecture can be split into three distinct domains: (i) a master domain; (ii) a controlled domain; and (iii) a network domain.

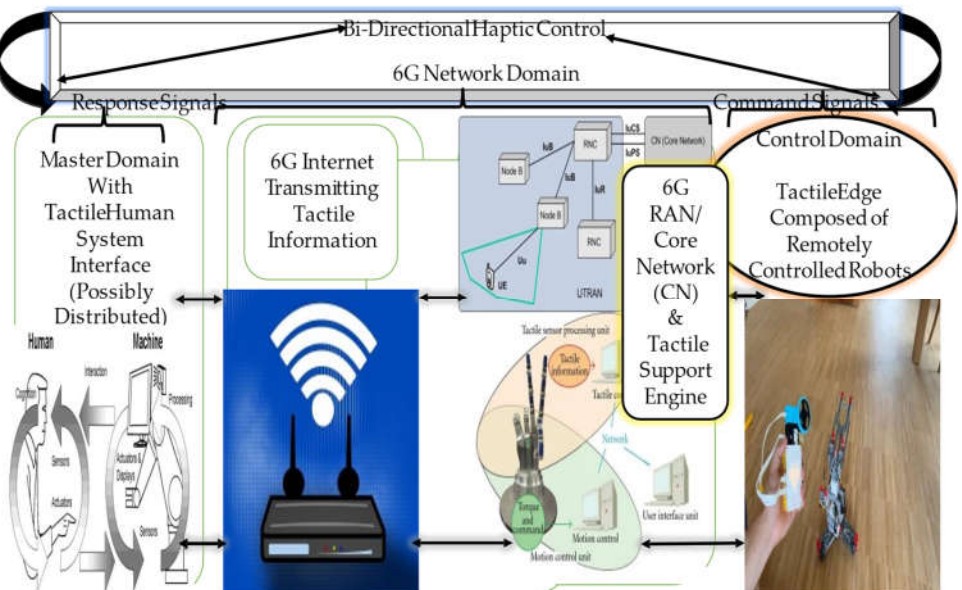

**Figure 10.** Functional representation of the 6G enabled Tactile Internet architecture.

### 5.2.1. The Master Domain

The master domain constitutes a human (operator) and a human–system interface, a haptic device, and has the provisioning for audio and visual feedback which converts the human input to tactile input through tactile coding methods, allowing the user to touch, feel, and manipulate objects in real and virtual environments. This primarily controls the operation of the controlled domain.

### 5.2.2. The Control Domain

The control domain includes a teleoperator (controlled robot) and interacts with objects in the remote environment. It is directly controlled by the master domain via various

command signals. The energy is transferred between the master and the controlled domains via command and feedback signals to achieve the control loop closure.

### 5.2.3. The Network Domain

The control domain facilitates the medium for full-duplex communication via 6G between the master and the controlled domains to attain the coupling between the humans and the remote environment.

### 5.2.4. RAN/CN

The 6G-driven communication architecture composed of RAN and CN meets the basic requirements to realize the 6G enabled Tactile Internet. To achieve this objective, the essential functions of the 6G RAN in the Tactile Internet ecosystem are as follows: (i) radio access technologies (RATs) such as mmWave, massive MIMO, and full-duplex; (ii) Tactile QoE/QoS for tactile applications in conjunction with M2M and smart grids applications; (iii) efficient packet delivery through reliable radio protocols and physical layer; and (iv) novel medium access control (MAC) techniques, where the air–interface conflicts are optimally controlled. The key functionalities of the 6G CN associated with the Tactile Internet are as follows: (i) dynamic application of QoS provisioning; (ii) edge–cloud access; and (iii) security.

### 5.2.5. End-to-End Latency

End-to-end latency of less than 1 ms remains the main challenge related to the system realization of a Tactile Internet. As the human reaction time is of the order of 1 ms, the P2M communication faces no cyber issues; therefore, it is a vital requirement for a reliable Tactile Internet system. The following are the factors that affect the end-to-end delay: (i) propagation delay; (ii) transmission delay; (iii) queuing delay; (iv) coding and decoding process; (v) routing process; and (vi) protocol stack optimizations. The Tactile Internet must handle the tactile data in the same manner as the conventional audio/visual information. Hence, the communication between smart sensors or devices for healthcare requires encoding mechanisms which provide transmission of tactile data over packet-switched networks.

### *5.3. Revolutionizing Healthcare with CDI*

### 5.3.1. CC Architecture for Cognitive Healthcare

Computing architecture based on massive data/CC system architecture includes network technologies (6G Tactile Internet, IoHE), data analytics, and cloud computing. The primary applications of the CC system, as shown in Figure 11, include cognitive healthcare, health monitoring, etc. Every layer in CC system architecture faces technological challenges and system needs. The development of the IoHE collects a variety of valuable information to bestow greater understanding and transmission of data, where it can provide an important source of information for the realization of CC. This system technology can process relevant information using intelligent computing techniques such as cloud computing, tiny ML, and data mining to make decisions. Since the emergence of tiny ML, cloud computing resources have provided tremendous benefits for advancing CC. The IoHE and cloud computing can present software- and hardware-based CC, while MDA provides novel ways to explore opportunities for data such as human massive data thinking.

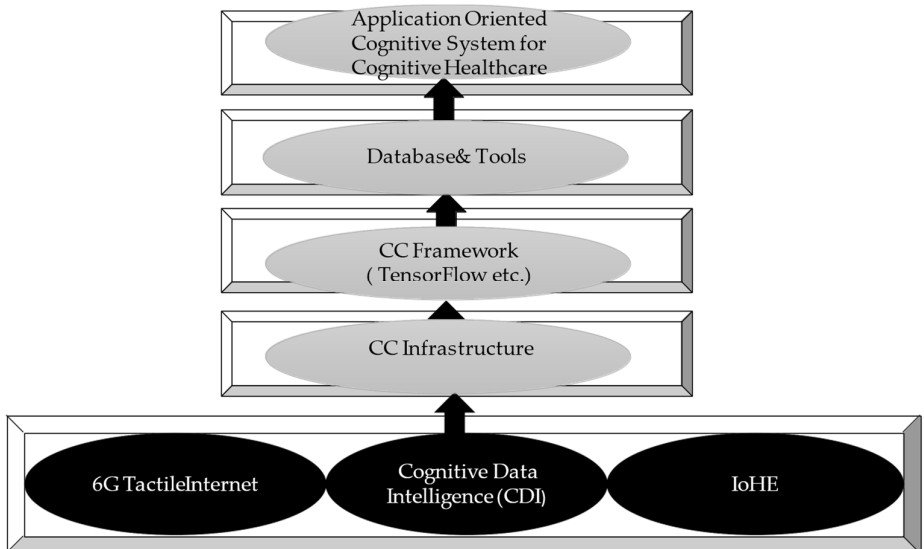

**Figure 11.** Cognitive system architecture for cognitive healthcare.

5.3.2. Linking CC, MDA, and Tiny ML for Healthcare Value Creation

CC is a key enabler of these human-centered smart systems and is a subdivision of AI, and anything that is cognitive is also AI. CC is also an AI-based system that enables it to interact with humans as a fellow human and interpret the contextual meaning to help the humans in decision-making. The mapping between features of massive data and cognitive computing and the features associated with the link between massive data and CC are shown in Figure 12.

CC can be used to process large volumes of data and comprises the concepts of observation, interpretation, evaluation, and decision (described below) that are mapped to five attributes of massive data: (i) volume; (ii) variety; (iii) veracity; (iv) velocity; and (v) value:

1. Observation is imperative in a CC system in which data aggregation, integration, and examination are done, hence, data volume must be available for observation.
2. Interpretation provides a better understanding and solving of complex problems in the presence of a variety of information sources, hence, variety indicates that data can be sourced in a variety of ways, such as IoT, social media, email, etc.
3. Evaluation is the natural ability of a human being to produce information, therefore, processing a massive amount of data needs evaluation in real time by the CC system. Velocity is a feature of big data, where speed is a vital requirement for production control and processing.
4. Decision feature is to make decisions by a CC system as per the analyzed data. Veracity defines quality prediction, uncertainty, and reliability of data. The value attribute indicates the valuation of massive data prior to it becoming knowledge creation.

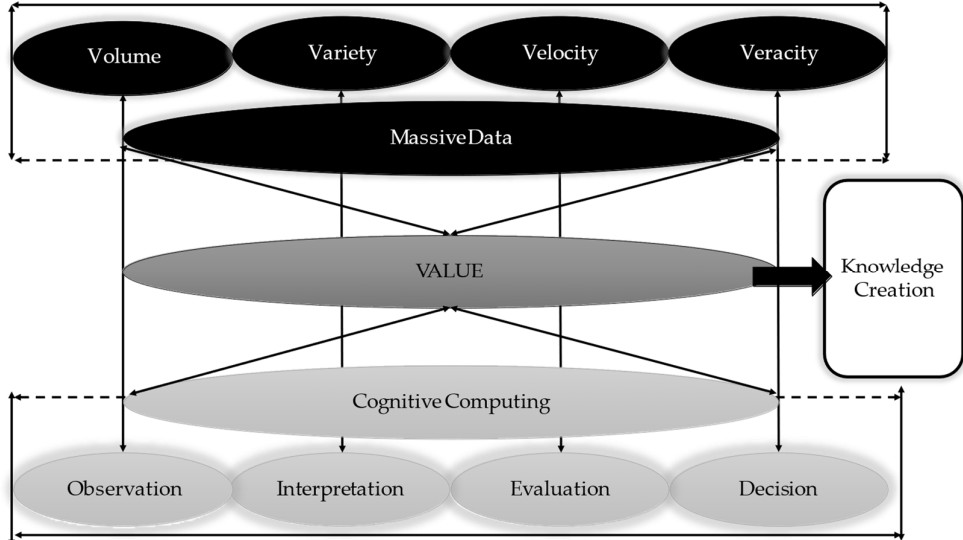

**Figure 12.** Linking between CC and massive data for knowledge creation.

Quantifying the health condition and the associated risk of an individual is one of the biggest innovations. The next phase of healthcare will bring new inventions into the picture which will not only increase the lifespan of individuals but will also predict diseases through eXR analysis. This revolution has just started, and the benefits of these technologies in the healthcare sector are immense. Technology innovation has been a natural phenomenon for many decades, and advancement of IoT and AI, especially tiny ML algorithms, has driven human-centered smart systems to higher-quality services such as smart healthcare. Analyzing massive data by humans is a lengthy process. IoT devices and connections will grow in the course of time. This exponential growth is leading to an explosion of data, and the enormous amount of data being produced on a continuous basis is termed "massive data". The concept of massive data has brought out many definitions by other researchers in the healthcare domain.

Healthcare organizations seek to analyze raw data, as they want to identify the trends for further profit maximization. Data analysis by humans can be time-consuming. Therefore, use of sophisticated CC can be applied to crunch the massive amount of data. The dire need to address the concerns of data deluge led to the emergence of MDA. Hence, MDA has gained significant importance, as it enables organizations to achieve a sustained competitive advantage.

AI and data analytics are not new concepts. The advancements in MDA and AI have enabled the early detection and prediction of such diseases. Tiny ML emerged as one of the most critical game-changing technologies and propels healthcare organizations to provide better security and a deeper understanding of the relationships and trends in inpatient data. Healthcare organizations are at the helm of ML and must use AI to remain competitive and secure in the IoHE healthcare platforms. When selecting an IoHE platform, organizations must understand which platform functions use tiny ML and IoT data, testing and validation protocols for ML, and how the ML updates. If a healthcare organization uses tiny ML for security, the organization must confirm if the patient data are used to retrain and update the ML's knowledge base to best reflect and improve the ML's trustworthiness.

The application of CDI (CC, MDA, and tiny ML) is going to reduce considerably the complexities and the uncertainties associated with the healthcare industry. With the mining of a massive amount of data availability and technologies, the world is witnessing the next phase of healthcare applications. With the availability of a vast amount of daily data and tiny ML, the algorithms bestow better results with higher accuracy and in less time

for the healthcare sector. Thus, the integration of CC, MDA, and AI, especially tiny ML, bestows the co-creation of value by the process of transferring data to information to knowledge to wisdom, as shown in Figure 13, that can help to better understand the complexity of data deluge.

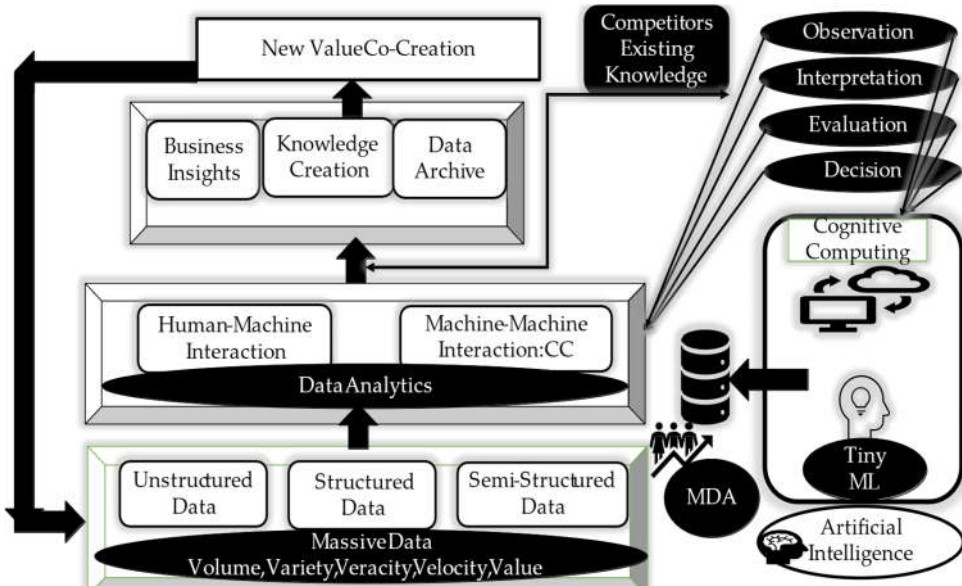

**Figure 13.** Linking CC, MDA, and AI for healthcare value creation.

## 6. The 6GCIoHE System Architecture

Research shows that there is no consensus on the 6GCIoHE system architecture. The three-layer architecture defines the main idea of the CIoHE, but it is not sufficient for researching the 6GCIoHE system. Healthcare applications have different requirements; therefore, the five-layer architecture is chosen to directly affect the application performance. Furthermore, when a healthcare project is implemented with cutting-edge technologies (6G Tactile Internet, CDI, IoHE) and broad application areas, the five-layer architecture is considered best. That is why we proposed the five-layer architecture. The five-layer 6GCIoHE architecture proposed is inspired by the layers of processing in the human brain and the ability of human beings to think, feel, remember, make decisions, and react to the physical environment. The proposed 6GCIoHE system architecture addresses the latency issues present in cloud-based solutions. This architecture also supports time-critical healthcare applications that require an emergency response and contains five layers, as shown in Figure 14, namely: (i) the sensing layer (perception layer) comprises the physical elements such as sensors, devices, and machines; (ii) the communication layer (transport layer) includes the communication systems such as 6G Tactile Internet; (iii) the processing layer (middleware layer) consists of communication protocols, cloud computing, and CDI software to analyze data; (iv) the application layer deals with intelligent applications; and (v) the business layer defines the business model.

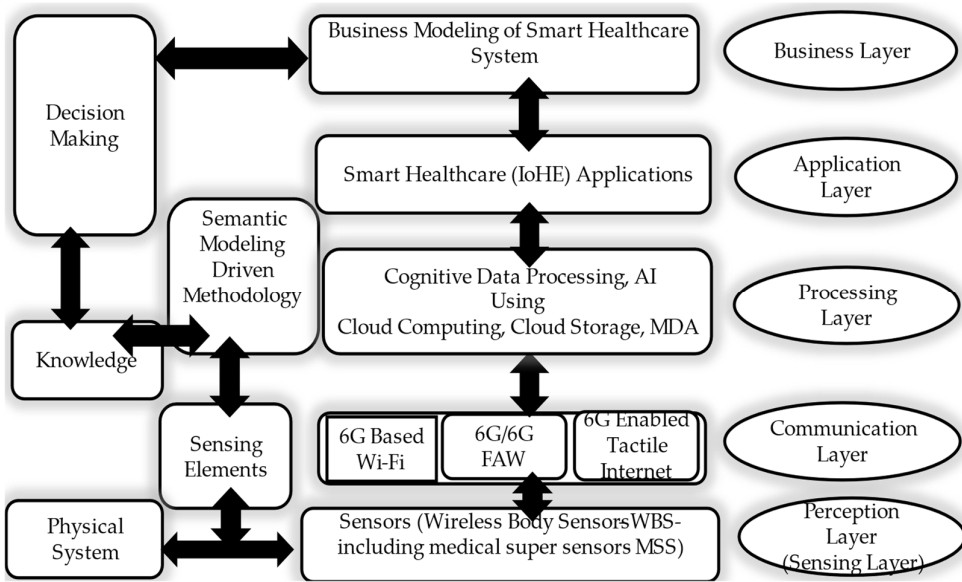

**Figure 14.** The 6GCIoHE system architecture.

To capture real time data concerning the physical world, the CIoHE phenomenon is fast gaining momentum in different application domains, especially in healthcare. Inspired by the human nervous system and cognitive abilities, a set of autonomic cognitive design patterns alleviate the design complexity of smart IoHE-based systems while taking into consideration CDI and scalability management. These patterns are articulated within a model-driven methodology that we proposed in Section 4 for developing a flexible cognitive monitoring system to manage a patient's health based on heterogeneous sensors.

The usage of the 6GCIoHE system technology in the healthcare domain is still in its nascent stage. As such, many challenges need to be addressed by the research communities and the industry. Some challenges and the existing solutions are discussed in the next section.

### 6.1. Sensing Layer

The sensing (perception) layer contains different sensors for monitoring the health parameters of patients and transmits the data to application devices. As a WBS, including MSS, senses data from the patient, it needs to be transmitted to the processing node via the communication network. Some of the key technologies useful for transferring data from the MSS to the closest processing node are 6G Tactile Internet or 6G-based fixed wireless access (FWA) or 6G Wi-Fi [1] (p. 15).

### 6.2. Transport Layer

The technology at the heart of communication for the transport layer in the IoHE architecture would be the 6G Tactile Internet that offers speed, accuracy, real time latency, energy conservation, and reliability among sensors and actuators close to a human body. The 6G Tactile Internet, 6G Wi-Fi, or 6G FAW can help to reduce healthcare costs and improve quality by using different sensors to read a patient's health parameters, thus improving the patient's quality of life.

### 6.3. Processing Layer

Cloud computing is a computing model that bestows a pool of configurable resources that can be accessed ubiquitously through the 6G enabled Tactile Internet to provide cloud services such as Software-as-a-Service, Platform-as-a-Service, Infrastructure-as-a-Service, and Database-as-a-Service. Cloud computing is one of the enabling technologies for IoHE

and offers significant benefits to healthcare organizations such as rapid elasticity, self-healing, and self-configuration. It is employed at the processing layer, also called the middleware layer, in the five-layer 6GCIoHE architecture. In the IoHE, the multitude of sensors generates a massive amount of data that can be stored and analyzed with the help of cloud infrastructure. Medical professionals can monitor a patient's health, whose information is collected through various sensors and is stored in the cloud. Even though the cloud provides many advantages, one of the key challenges is latency. The process of data collection and analysis by the sensors is done through the Internet and needs to reach physicians. This process takes a significant amount of time, which might not be suitable for emergency healthcare services. The ideal solution to reduce time delay is the latency offered by 6G enabled Tactile Internet, or 6G Wi-Fi, or 6G FAW networks.

### 6.4. Application Layer

In this layer, state-of-the-art applications of the 6GCIoHE system-related technologies in the healthcare domain are presented. Healthcare applications have different requirements; therefore, the five-layer architecture is chosen to directly affect the application performance. Applications with similar themes and scope are therefore grouped into categories as follows: (i) real time monitoring and alerts generation; (ii) telemedicine; (iii) chronic disease detection and prevention; (iv) home and elderly healthcare. Each of these healthcare application categories and the role of the 6GCIoHE system and the associated technologies are as follows: (i) through the IoHE, WBS can be deployed on a human body to measure different parameters, and data can be analyzed for prescribing the necessary medication to the patients in emergencies; and (ii) an e-health system is used for continuous monitoring of ECG, temperature, foot pressure, and heart rate.

### 6.5. Business Modeling Layer Value Co-Creation through Business Models

The business model concepts underlying the disease have their precedents in technology-based businesses that closely mirror those seen in the ICT industry. The evolution transformed through product innovations that compelled it to reinvent its business model, shifting to provide services and solutions for smart healthcare. As with healthcare today, ICT a generation ago was part of a classically maturing market with disruptive innovation. One must re-imagine business models to find ways to innovate at scale and with speed as a differentiator for improving healthcare and earning superior returns.

## 7. 6GCIoHE System for Healthcare Applications and Challenges

### 7.1. Applications

Currently, the amount of healthcare data is doubling every 73 days. CC is defined as a system that learn at scale, allowing it to interact with humans more naturally. CC can understand the natural language and can process massive amounts of data to comprehend and learn from them, helping healthcare providers with care plan enhancements. Cognitive systems bestow advice to individual patients and caregivers by developing deep domain insights and adducing this information to patients in a usable, natural, and timely manner. The ICT infrastructure needs to be malleable enough to harmonize applications with distributed devices and fast-track digital applications with IoHE systems to ensure data protection and powerful security. There are several new IoHE solutions that help doctors and patients communicate without needing in-person visits to get better treatment and save money [33] (p. 3).

The remote symptom monitoring has improved for both patient outcomes and experience versus patients who had routine in-person visits. The symptom monitoring can also allow for real time adjustments to treatment besides simplifying data collection and transmission between a patient and doctor.

The 6GCIoHE system can address the following wide-reaching capabilities used across disciplines, sectors, or treatment efforts and medical conditions: (i) symptom

monitoring/tracking; (ii) medicine adherence; (iii) ingestible sensors; (iv) hygiene monitoring; (v) body scanning; (vi) smart labs; (vii) diabetes; (viii) Parkinsons; (ix) asthma; (x) cognitive/mental health; and (xi) Alzheimer's.

The future of population health management (PHM), as shown in Figure 15, will be trussed to CC, which converts unstructured data into structured data by utilizing massive parallel processing and AI to search the healthcare literature.

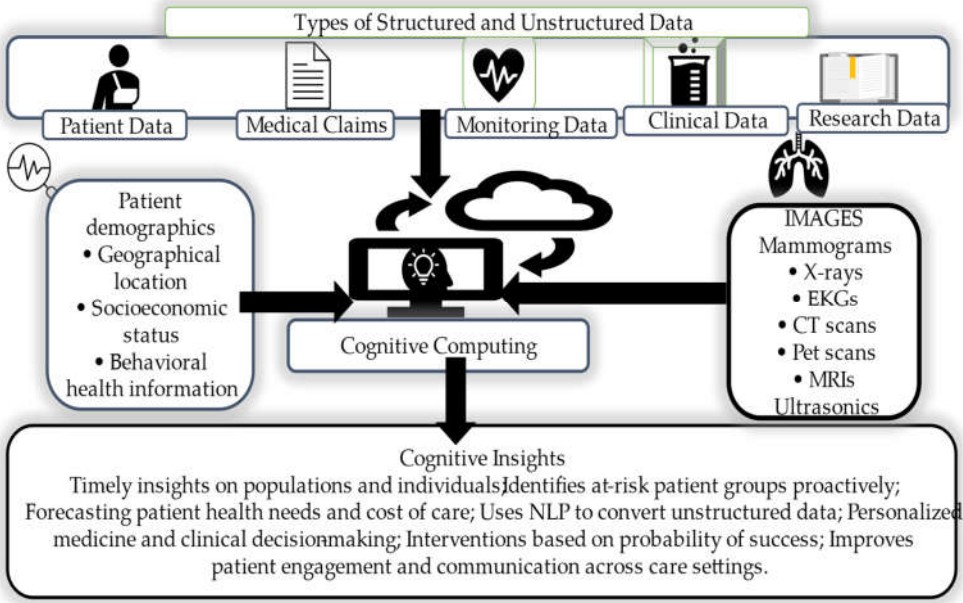

**Figure 15.** Cognitive computing for population health management.

A new era is emerging, from digital to cognitive, for healthcare applications where clinicians can collaborate with cognitive computing systems to improve healthcare. With ever-increasing healthcare information from connected medical devices, personal fitness trackers, implants, and other sensors that collect a massive amount of real time data, one person in their lifetime is likely to create more than 1 million gigabytes of health-related data.

A smart monitoring system is essential to manage patients' health evolution based on WBS, including examples such as: (i) in the context of managing diabetes, WBS can be used to measure the "Blood Sugar". Thus, the cognitive monitoring management pattern identifies the interactions among the WBS and the physicians for visualization and anomaly detection; (ii) the semantic knowledge pattern manages data heterogeneity and enables the collaboration among the WBS and the physicians; and (iii) the massive data stream detection pattern manages data heterogeneity and velocity.

### 7.2. Challenges

At present, the healthcare sector is facing numerous challenges, however, emerging cutting edge digital technology, such as the 6GCIoHE system, can assist to stay ahead of the curve and lend the following solutions: (i) healthcare mobile and web applications driven by 6G Tactile Internet can offer powerful virtual consultations; (ii) social networking applications can be used by healthcare professionals across the globe via the 6GCIoHE system; (iii) innovation in the Medicare EHR system can make reimbursements easily; (iv) data management systems can automatically oversee the inventory; (v) cloud plus data analysis capabilities enables efficient management; (vi) block-chain based database solutions need to be implemented; (vii) prescription based online medical stores need to be established; (viii) remote surgery is one of the typical application scenarios for the

6GCIoHE system; and (ix) the 6G Tactile Internet communication system is also a typical application scenario for emotional remedy purposes.

The following principles offer stakeholders the opportunity to address the IoHE security challenges.

(i) Security implementation at the design stage: in smart healthcare, it becomes a reality of an innovative concept that offers the best service to patients. The IoHE takes new challenges in the healthcare field to create excellent support systems for patients and medical professionals. Comprehension of consequences for the failure of a device enables healthcare stakeholders to suggest informed risk-based security decisions. Hence, one should design with system and operational disruption in mind so that the failure does not lead to greater systemic disruption, e.g., WBS is used to sense and capture the data related to patient health/disease via the Internet, and devices display continuous process monitoring; (ii) promotion of security updates and vulnerability management by leveraging cryptographic integrity and authenticity protections to address vulnerabilities via software updates and develop automated mechanisms; (iii) develop proven security practices enabling security by default through unique, impossible to hack usernames and passwords. Failure to execute adequate security measures could be damaging in terms of reputational costs and financial costs; (iv) security measures prioritization must be incorporated based on potential influence. The awareness of a device's intended use and environment helps stakeholders to consider the technical characteristics of the IoHE device and the security measures that may be essential. The mitigation planning and analysis should help prioritize decisions to execute additional security measures. Identify and authenticate the devices connected to the network; (v) propagate transparency across the IoHE spectrum: stakeholders should include vendors and suppliers in the risk assessment process to create transparency, enabling an increase of awareness of potential third-party vulnerabilities and promoting trust. Development of a software bill of materials that can be used as a means of building shared trust among the stakeholders should be realized. Furthermore, it can serve as a valuable tool in the IoHE ecosystem to manage risk and patch any vulnerabilities; and (vi) connectivity should be done carefully regarding nature and the purpose of connections to enable healthcare stakeholder decisions. One should build in controls to allow stakeholders to disable network connections when needed or desired to enable selective connectivity.

## 8. Toward A Theoretical Framework

### 8.1. Theory's Focal Phenomena

Theories offer an illustration of one's notion of a subset of real global phenomena. The high-quality theory is thought to enhance the knowledge of the researcher and other scholars' know-how within the theory's domain. It additionally served to enhance practitioners' abilities to perform efficiently and efficaciously within a theoretical framework. A theory makes novel contributions to a discipline in the following ways: (i) prior theories are not covered by a theory's focal phenomena; (ii) a theory is probably taken into consideration as novel because it frames or conceives existing, well-known focal phenomena in new ways; and (iii) a theory's novelty arises due to essential modifications it makes to an existing theory, more precisely, adding or deleting constructs or specifying the boundary of the theory more concisely.

Thus, the theoretical framework proposed in this paper falls under a theory's focal phenomena that were not included in prior theories.

### 8.2. Security, Privacy, and Familiarity Affecting Trust in the 6GCIoHE System

To date, no study investigated the approaches where the 6GCIoHE system security, privacy, and familiarity can affect trust and, in turn, how trust can affect risk perception and attitudes toward use in healthcare applications. No research considered ways in which risk perception can mediate to bolster the relationship between trust and users'

attitudes toward using the 6GCIoHE system technology at the heart of communication for the transport layer in the IoHE architecture. This would be the 6G Tactile Internet that offers speed, accuracy, real time latency, energy conservation, and reliability among sensors and actuators close to a human body. The 6G Tactile Internet, 6G Wi-Fi, or 6G FAW can reduce healthcare costs and improve its quality by using different sensors to read a patient's health parameters, thus improving the patient's quality of life.

Cloud computing is one of the enabling technologies for IoHE that offers significant benefits to healthcare organizations such as rapid elasticity, self-healing, and self-configuration. It is employed at the processing layer, also called the middleware layer, in the five-layer 6GCIoHE architecture. In the IoHE, the multitude of sensors generates a massive amount of data that can be stored and analyzed with the help of a cloud infrastructure. Medical professionals can monitor a patient's health, whose information is collected through various sensors and is stored in the cloud. Even though the cloud provides many advantages, one of the key challenges is latency. The process of data collection and analysis by the sensors is performed through the Internet and should reach physicians. This process takes a significant amount of time, which is not suitable for emergency healthcare services. The ideal solution to reduce the time delay is the latency offered by 6G enabled Tactile Internet, 6G Wi-Fi, or 6G FAW networks. In this layer, state-of-the-art applications of 6GCIoHE system-related technologies in the healthcare domain are presented.

The business model concepts underlying the healthcare ecosystem have their precedents in technology-based businesses that closely mirror those seen in the ICT industry. As with healthcare today, ICT a generation ago was part of a classically maturing market with disruptive innovation. One must re-imagine business models to find ways to innovate at scale and with speed as a differentiator for improving healthcare business models and earning superior returns. Consequently, this study sheds insights on developing a novel framework to measure the causes and the effects of contingency factors and how they can influence users' attitudes toward using the 6GCIoHE systems. Security, privacy, and familiarity factors are discussed below in more detail to pave the way for developing the theoretical framework.

### 8.2.1. Security

Researchers reveal significant insecurity in the IoHE devices. The IoHE wireless devices pose many security challenges such as intrusion, denial of service, forgery, or heterogeneous network threats. Hackers can use ransom ware to target vulnerabilities in the Windows operating system to prevent healthcare professionals from accessing affected devices.

Security is defined as the protection of software and hardware from misuse, malfunction, unauthorized access, damage, disruption, and misdirection. Security is important to the IoHE applications because of sensitive health data privacy; hence, the following security solutions are imperative to protect data from attacks: (i) access control is a key step in protecting IoHE applications and health data, therefore, well-designed access control and strong access management must be implemented to ensure healthcare data security and privacy. Through access control systems, an organization can restrict and monitor the use of critical data and protect privacy and security. Furthermore, healthcare stakeholders should have training and awareness of information security to provide security of IoHE applications and health data; (ii) the IoHE devices collect data from various environments, hence, physical security is paramount for the IoHE devices. Physical security of IoT health devices and health data involves protection against environmental threats, accidents, physical sabotage, and theft; (iii) IoHE devices should have replacement devices for protecting physical attacks so that the IoHE devices continue collecting and transferring data without interruption; (iv) network security, such as 6G Tactile Internet, is an important issue for IoHE devices and applications. All IoT devices connect to networks and communicate with each other over a network. Firewalls, filtering structures, internet protocol

(IP) security, and secure sockets layer/transport layer security should be used to ensure network security.

Ethical hacking can identify security vulnerabilities and risks for an organization. Moreover, most of the devices are designed without security, making them easy goals for safety breaches. By using the IoHE hacking tools, one can secure the system and the infrastructure within the organization. This process helps to create an ideal strategy that effectively and efficiently incorporates tools to identify and resolve security vulnerabilities.

Ways to secure the IoHE devices include: (i) developing an exhaustive map of healthcare assets; (ii) following the best security practices; (iii) effective authentication implementation; (iv) insulating devices that do not have a built-in control; and (v) usage of appropriate tools.

### 8.2.2. Privacy

Privacy has many definitions in the literature. Privacy is an important topic for information security in the healthcare system in the world. In this study, privacy is defined as freedom from unauthorized intrusion and an important challenge in the IoHE environment due to the availability of sensory devices and the speed and the volume of information flow. Any compromise of privacy leads to problems such as eavesdropping, unauthorized access to or alteration or destruction of information, hacking, identity theft, forgery, and social engineering.

Data privacy is a fundamental issue for IoE health devices and applications because of the ubiquitous character of the IoE environment. IoE health devices connect to each other for data transmission and a strong encryption algorithm that is used for data to be encrypted over a secure network such as 6G Tactile Internet. Healthcare data are collected from the IoE devices via remote access mechanisms which have some challenges regarding privacy and security. Data collected by the sensor are transmitted to the database or the cloud over a network such as 6G Tactile Internet. Since healthcare data include essential significant information, the world's hackers want to capture health data. Therefore, the privacy of data must be protected. Some health organizations are reluctant to adopt the IoE because of fears of privacy compromises, particularly in cases that involve medical data in which maintaining privacy and anonymity of the user is of the utmost importance due to legal and statutory requirements, which in turn affects trust to adopt the IoE in the healthcare domain.

The IoHE application must have a "need-to-know" principle for authorization management. The stakeholders should have enough information about procedures, guidelines, standards, and policies related to data security and privacy. All IoHE devices, IoHE applications, and network components log must be collected with central log management systems. Besides, central log management or security information and event management must have auditing to ensure security. Undesirable events must be reported to a security team quickly to interfere with unwanted events. Central logs management or security information and event management must have strong authentication and authorization to monitor the audit log. The log should be checked continuously.

### 8.2.3. Familiarity

Familiarity means good knowledge of some fact or knowledge primarily based on preceding interactions. Scholars verified the significance of familiarity and trust in an e-commerce potential and argue that familiarity has an indirect positive influence on the intention to undertake endorsed agents.

### 8.3. Trust in the IoHE

Researchers use trust in the IoHE as a conventional trust framework to discuss security in cell networks as the important anchor of monitoring device behaviors, identifying devices, connection protocols, and the related procedures to devices. Security measures at

the device level could be adopted to enhance security. At the network level, security could be improved by using point-to-point encryption techniques based on cryptography algorithms, message integrity verification strategies, and trusted routing mechanisms. Security measures to prevent data security and privacy are required to be adopted at the cloud level, and suitable training concerning awareness is needed at a human level.

Researchers proffered the following to attain trust in the IoHE: (i) design of a trust framework to be included within the development of any IoHE entity; (ii) advanced dynamic protocol for trust management which enables IoHE systems to deal with misbehaving nodes whose status or conduct might change dynamically; (iii) offered an extensible trust model that was seamlessly integrated into the IoHE ontology and focused on the IoHE-trust modeling, an ontology for fuzzy semantics reusing existing trust models; (iv) designed meta-models for contractors by defining privacy and quality of context conventions independently of those of the users and the creators for the independent management of privacy in the IoHE; and (v) established a proper trust–management control mechanism primarily based on the architecture modeling of the IoHE.

However, the dearth of consideration of trust in relation to the quality of experience (QoE) is also seen as a shortcoming and is one of the significant challenges concerning trust in the IoT establishment of remote IoHE devices.

Devices on the Internet must be trusted to ensure privacy and security. Trust management is important for IoHE devices and applications to provide security and privacy of data. Hackers could connect devices to manipulate data in the IoHE applications. Data collection trust is a serious issue because a massive amount of data is collected from devices and are utilized by IoHE applications, which need make the right decisions about patients to improve the quality of healthcare.

### 8.4. Risk Perception

"Risk perception" is described as the subjective judgment that people make about characteristics and severity of a risk. Researchers investigated a well-known perceived risk scale to undertake the IoHE applications. Research shows a correlation between user's risk perception related to the IoHE device being able to prevent one from being hacked (one of the risk components). Researchers also found that (i) respondents are not in consensus concerning the perception of risk; (ii) perceived risk influenced consumers' online behavior; (iii) perceived risk is a major obstacle in the IoHE adoption; (iv) analysis of risk perception among users of smart devices linked to the IoT at home and found that risk perception is associated with knowledge of and anxiety regarding the devices; (v) risk perception also is a key factor in determining the IoHE adoption; and (vi) trust is a crucial factor in adopting the IoHE satisfactorily and in regard to expected transaction outcomes.

### 8.5. Attitude towards Using the 6GCIoHE System

Attitude can be described as an experience or opinion regarding something or someone or a way of behaving caused by something or someone. Scholars discovered the following associated with mindsets in the healthcare domain: (i) doctors embrace notable attitudes towards the IoHE based medical devices, which meant that they are privy to and are prepared to adopt technology and ascribe an exceptional quality to the information transmitted; (ii) maximum customers of the IoHE in healthcare support progressive perspectives concerning valuable features and desire solutions in regions which include inventory or material tracking, identification, and authentication that could make healthcare offerings more effective, convenient, and safe; and (iii) even patients consider favorable perspectives employing the IoHE devices.

*8.6. Building Blocks of 6GCIoHE System Theoretical Framework*

Based on Sections 2–7 and Sections 8.1–8.5 above, the advancements of the proffered theoretical framework focused on the building blocks and the practices, as shown in Figure 16, are the following:

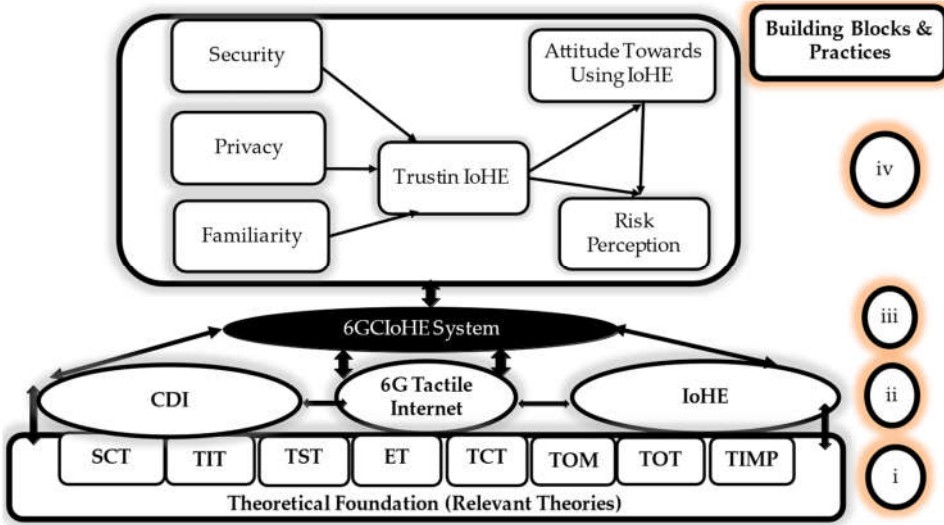

**Figure 16.** The 6GCIoHE system theoretical framework.

(i) The design of the 6GCIoHE system paradigm based on the theoretical foundation which includes the relevant theories; (ii) the 6GCIoHE architecture to exploit the challenges and the opportunities in the cognitive healthcare applications; (iii) the collaboration among the healthcare stakeholders within the 6GCIoHE system value chain with the interactive actions between 6G Tactile Internet, CDI, and IoHE applications to achieve optimum performance in the healthcare sector; (iv) the security, the privacy, and the familiarity affecting trust in the IoHE implementation; (iv) the risk perception as an important factor in determining the IoHE adoption; and (v) the risk perception mediating the relationship between trust in the IoHE and users' attitude towards using the IoHE.

The proffered theoretical framework would assist in understanding, evaluating, and accessing the emerging 6GCIoHE system for millions of people for higher overall performance of the healthcare applications.

*8.7. Theoretical Framework Contributions*

An overall contribution of our framework is the establishment of theory for the 6GCIoHE system adoption and implementation, which was previously non-existent in the literature.

The additional contributions of the framework components as related to the healthcare industry and research are as follows: (i) even though the 6GCIoHE system involves M2M communication, the human component cannot be ignored from the model; and (ii) the 6GCIoHE system, a multilayer stack of technologies, contributes new knowledge regarding privacy and security to gain user trust.

**9. Consolidated Lessons Learned and Future Research Direction**

Currently, the radio channel above 140 GHz and future gigabyte Wi-Fi with IoT is little understood. Hence, 6G wireless and beyond should be pursued with cutting edge work in propagation measurements and modeling above 140 GHz. One important new technology being researched is 6G cell-free massive MIMO, a radical network architecture

with the goal of eliminating interference between cells with higher-frequency spectrum bands, including THz bands.

The IoHE network demands massive and smart connectivity, huge bandwidth, and real time latency with an ultra-high data rate for a quality healthcare experience. The emerging 6G communication system is expected to provide intelligent IoHE services ubiquitously at any time to improve the quality of life of human beings.

The proliferation of eHealth, including cognitive healthcare services, challenges the ability to meet their stringent quality of service (QoS) requirements, i.e., continuous connection availability (99.99999% reliability), extremely low latency (sub-ms), and mobility support. The swanky intelligence of emerging 6G networks combined with a wider spectrum will guarantee key performance indicators (KPIs) and gains in spectral efficiency by up to 10 times. The 6G enabled Tactile Internet and its aggregation with multidimensional techniques offers a newly distributed security paradigm in the context of intelligent IoHE applications with less than 1 ms latency for connecting homes to hospitals, healthcare people, medical devices, and hospital infrastructure, etc., and 6G will also eliminate time and space barriers through remote surgery.

Massive data management is a challenge for healthcare researchers for many reasons, such as huge volume, high production velocity, different sources, data quality, resource reliability, etc. CC is in its nascent stages, and the success of using CC technologies for MDA in cognitive healthcare can be interesting for future studies.

The ever-growing diffusion of WBS and IoHE devices is heavily changing the way healthcare is approached worldwide. The 6GCIoHE system architecture needs to be exploited further to make smart healthcare systems capable of supporting real time applications when processing a massive amount of data produced by WBS networks.

The tiny ML concept leads data scientists to develop artificial neural networks and deep learning algorithms through biomimicry. Such technologies in the context of practical clinical research show how tiny ML can act as a tool to support and amplify human cognitive functions for physicians delivering care to increasingly complex patients.

Currently, the IoE connections are disparate and clunky, and connecting devices do not create value like connecting people. A central problem with IoE's current architecture is that users are forced to trust platforms.

The literature on the cognitive healthcare topic lacks a holistic view of the current state of research and application. Future research should consider the philosophical perspectives underlying theoretical accounts to ensure that their techniques are consistent with the cognitive-developmental models.

## 10. Recommendations

Today's avant-garde technology is evolving in the age of unparalleled academic research. Assuming incantation performs the role of the supernatural, the avant-garde technology should play the "instinctive" position. "Telekinesis" [41] is the potential to perform real-time physical action over a remote object without having a physical connection, specifically Tactile Internet and IoHE. The Tactile Internet will permit remote manipulation of objects, as in remote surgery, while imparting the feeling that the remote object is at the fingertips because of the extremely low latency of the 6G connection to offer that real time excitement. The integration of these technologies can augment the tactile capabilities of the individual toward distances that are beyond the usual human reach. Hence, we recommend the possibility of redesigning the manner to engage within a smart healthcare system for diverse factors of living.

Human-centric mobile communications must be one of the most important applications of 6G enabled Tactile Internet and consequently desires careful attention by the wireless research community. Data rate requirements of one terabit per second, end-to-end reliability with latency much less than one millisecond, and higher security, secrecy, and privacy must be key features of 6G Tactile Internet and must address application types: (i) MTRLLC; (ii) mURLLC; (iii) HCS; and (iv) MPS [1] (p. 15).

Mobile operators globally are accelerating their 5G RAN investments. Open RAN will capture a certain percentage of the total RAN market. Many operators are keen to accelerate the pace of open RAN developments and enable these open interface-based systems to achieve performance and functional parity with "purpose-built" systems. We recommend open RAN to become a fundamental part of the 6G mobile system solutions.

The 6GCIoHE system ought to meet healthcare utility necessities on latency, reliability, connectivity density, and gaining KPIs commonly 50 to 100 times higher than 5G. CDI must gain collating pathogenic and affected person information consisting of affected person history, pleasant practices, and diagnostic gear to investigate huge quantities of information to offer advice primarily based on real time desires describing the four salient characteristics: (i) understand; (ii) reason; (iii) learn; and (iv) engage. The IoHE must re-invent the healthcare industry at three levels: (i) a business process with digital technology that drives to improve products, services, and processes as well as customer and healthcare constituent experiences; (ii) a business model where digital products and the process should drive new ways of doing business for the healthcare industry with transformational changes as digitalization re-invents at the business model level; and (iii) a business moment where digital re-invention is created through the need to compete with unprecedented business velocity and agility, specifically the "business moment" would provide relevant and efficient ethical and security measures. Stronger cooperation is needed between the relevant stakeholders involved in the IoHE solutions for ethical and secure use of the IoHE, which is as follows:

1. User consciousness and trust should be built.
2. Consumers should know their rights to secure medical information.
3. To guard medical data and information of the population, the government must establish appropriate standards in terms of information security and apply them to all medical institutions.
4. From the manufacturers' perspective, a reputable system for IoHE devices should be provided for qualitative and objective benchmarks for trustworthiness.

We also recommend the following for the healthcare industry to achieve safe and secure connectivity: (i) manage security at every level of the 6GCIoHE system; (ii) protect the identity of objects and users; (iii) execute multi-factor authentication; and (iv) protect identities and not gateways.

## 11. Conclusions

This research satisfactorily performed the specific goals as stated in the introduction Section 1. A novel theoretical framework was advanced based on the perspectives outlined in Sections 2–7 and Sections 8.1–8.5. The key findings consist of proof of the 6GCIoHE system paradigm and associated dialog on the five-layer architecture and its applications, challenges, and beneficial effects on cognitive healthcare. The emerging 6G Tactile Internet is anticipated to be deployed on or after 2030 and shall facilitate the provisioning of tiny ML-as-a-Service to end-users via pervasive intelligence applied to enforce exceedingly efficient network transmission, optimization, control, and management of valuable resources. The use of cognitive systems must be to minimize the deficiencies of the MDA concerns. The research bestowed unique insights within the discipline of CDI, especially on the traits of CC that were mapped with the perspectives of massive data, specifically volume, variety, veracity, velocity, and value for knowledge creation. Inspired by the aid of the human cognitive process, we presented a comprehensive definition for the 6GCIoHE system paradigm security solutions to mitigate potential security threats in healthcare applications.

**Author Contributions:** Conceptualization, P.K.P.; Data curation, P.K.P.; Formal analysis, P.K.P.; Investigation, P.K.P.; Methodology, P.K.P.; Project administration, P.K.P.; Resources, P.K.P.; Software, P.K.P.; Supervision, F.C.-S.; Validation, P.K.P.; Visualization, P.K.P.; Writing—original draft, P.K.P.;

Writing—review & editing, P.K.P. Both authors have read and agreed to the published version of the manuscript.

**Funding:** This research received no external funding.

**Institutional Review Board Statement:** Not applicable.

**Informed Consent Statement:** Not applicable.

**Conflicts of Interest:** The authors declare no conflict of interest.

## Appendix A

### Acronym List

| | |
|---|---|
| AI | Artificial Intelligence |
| AR | Augmented Reality |
| CC | Cognitive Computing |
| CDI | Cognitive Data Intelligence |
| CH | Cognitive Healthcare |
| CIoHE | Cognitive Internet of Healthcare Everything |
| CN | Core Network |
| CT | Cognitive Technology |
| EHR | Electronic Health Records |
| ET | Ethical Theory |
| eXR | eXtended Reality |
| 5G | Fifth Generation Mobile System |
| FWA | Fixed Wireless Access |
| GUIMS | Global Ubiquitous Intelligent Mobile Society |
| HCS | Human Centric Services |
| ICS | Industrial Control System |
| ICT | Information Communication Technology |
| IoT | Internet of Things |
| IoE | Internet of Everything |
| IoHE | Internet of Healthcare Everything |
| IoMT | Internet of Medical Things |
| IS | Information System |
| KPI | Key Performance Indicators |
| M2M | Machine to Machine |
| MDA | Massive Data Analytics |
| MIMO | Multiple-input multiple-output |
| ML | Machine Learning |
| mURLLC | Massive Ultra-Low Latency Communications |
| MPS | Multi-Purpose Services |
| MR | Mixed Reality |
| MSS | Medical Super Sensors |
| MTRLLC | Mobile Teraband Reliable Low Latency Communications |
| NFV | Network Function Virtualization |
| PHM | Population Health Management |
| P2P | People-to-People |
| P2M | People-to-Machine |
| QAM | Quadrature Amplitude Modulation |
| QoE | Quality of Experience |
| QoS | Quality of Service |
| RAN | Radio Access Network |
| RAT | Radio Access Technologies |

| | |
|---|---|
| SCT | Social Cognitive Theory |
| SDN | Software Defined Network |
| 6G | Sixth Generation Mobile System |
| TF | Theoretical Framework |
| TCT | The Communications Theory |
| TI | Tactile Internet |
| TIMP | Theories of Inference and the Mathematics of Probability |
| Tiny ML | Tiny Machine Learning |
| TOM | Theory of Mind |
| TOT | Theory of Transparency |
| TST | The Systems Theory |
| URLLC | Ultra-Reliable and Low-Latency Communication |
| VNF | Virtual Network Function |
| VR | Virtual Reality |
| WBS | Wireless Body Sensors |

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
