# Peer review of "6G Enabled Tactile Internet and Cognitive Internet of Healthcare Everything: Towards a Theoretical Framework"

_asi, doi:10.3390/asi4030066_

Round 1

Reviewer 1 Report

i am grateful for the opportunity to review your manuscript.

i offer the following for your consideration:

  • there are instances in your manuscript when the linguistic expressions are flowery and non-academic, examples are "scant", "bequeath" and "bestow". the words by themselves would not stand out so much (though some are somewhat archaic) but for the fact that they occur in sentences which are grammatically problematic, such as "this study fills the scant and bequeath ... new knowledge..." (here, "scant" is an adjective and there is no noun).
  • you may wish to have your manuscript proof-read by a native English speaker, this will help spot errors such as "mimes" for "mimics".
  • in the first paragraph of your manuscript, you say that "scholars are inspired to pursue..." and then you proceed to cite your own paper. self-citation in and of itself is fine (if it is not excessive) but here you refer to yourselves in the third person ("scholars are inspired"), when - in actuality - it is you yourselves who are inspired.
  • there are some sub-section headings which are missing, specifically, the headings for sub-sections 3.5.4, 5.2.4, 5.2.5, and 5.3.1. the tab indentation of these sub-sections also does not follow that of other sub-sections.
  • i am not clear about the purpose of sub-section 4.1 'use case with examples'. might you wish to consider shifting it further down to section 7 '6GCIoHE system for healthcare applications and challenges'?
  • sub-section heading 5.2.3 'network domain' is not formatted with Initial Caps (to align with the others).
  • in sub-section 7.2 'challenges' you introduce six principles. you may wish to consider elaborating more on how you derive them. presently they do not sit very comfortably in the manuscript because (to my reading) nothing in the preceding sections builds explicitly to these six principles.
  • in sub-section 8.1 'theory's focal phenomena' you write "so far, the development of new theories were unnoticed within the discipline". first, i am not clear what exactly is meant by theories being unnoticed. second, even if i were to be clear, this comes across as a sweeping generalisation.
  • in sub-section 8.2 'security, privacy...', you identify a gap, and elaborate on this gap in the entire first paragraph. i feel this is useful exposition and should be brought forward in your manuscript, say to sub-section 1.1 on 'problem formulation'. that is where your articulation of the gap would make (more) sense.
  • still in sub-section 8.2, your third paragraph begins "the business model concepts underlying the disease". here again, it is not clear to me what you are trying to say. to my mind, i do not understand how a business model may underlie a disease.
  • you end sub-section 8.2 with a colon ( : ), and you do so because the subsequent sub-sub-sections are a list. you may wish to consult with the Editor on the appropriate punctuation. i personally feel a full-stop / period ( . ) would be acceptable and appropriate.

Author Response

RESPONSE TO REVIEWER 1 REPORT

Appreciate your effort for reviewing the manuscript.

Proof reading including grammar/spelling was done using the following tools (a, b, c) and manually. The entire manuscript analysis ( originality, spelling, grammar, word choice, style, and vocabulary), was done using the tools (d, e, f, g) mentioned below:

  1. Free, Powerful English Grammar Checker | SCRIBENS
  2. https://smallseotools.com/grammar-checker/
  3. https://cloud.trinka.ai/editor/main/02f66b9d-73a3-46ab-ae85-13f626ad72bd
  4. https://smallseotools.com/plagiarism-checker/
  5. https://www.check-plagiarism.com/article-rewriter
  6. https://www.paperrater.com/plagiarism_checker
  7. https://www.paperrater.com/free_paper_grader

Your comments are answered (in blue bold color) point by point below:

1.There are instances in your manuscript when the linguistic expressions are flowery and non-academic, examples are "scant", "bequeath" and "bestow". the words by themselves would not stand out so much (though some are somewhat archaic) but for the fact that they occur in sentences which are grammatically problematic, such as "this study fills the scant and bequeath ... new knowledge..." (here, "scant" is an adjective and there is no noun).

Response: Corrected- Lines 90 to 92

2.You may wish to have your manuscript proof-read by a native English speaker; this will help spot errors such as "mimes" for "mimics".

Response: Appreciate your suggestion. Proofreading was done manually and using the tools mentioned above.

3.In the first paragraph of your manuscript, you say that "scholars are inspired to pursue..." and then you proceed to cite your own paper. self-citation in and of itself is fine (if it is not excessive) but here you refer to yourselves in the third person ("scholars are inspired"), when - in it is you yourselves who are inspired.

Response: Corrected: Lines 30 to 33.

4.There are some sub-section headings which are missing, specifically, the headings for sub-sections 3.5.4, 5.2.4, 5.2.5, and 5.3.1. the tab indentation of these sub-sections also does not follow that of other sub-sections.

Response: The sub-sections 3.5.1, 3.5.2, 3.5.3, are corrected. 3.5.4 removed. Also, sub-sections 3.6.1 to 3.6.8, 5.2.1 to 5.2.5,5.3.1,5.3.3, appropriately corrected.

Formatting and titles are corrected throughout the article.

5. I am not clear about the purpose of sub-section 4.1 'use case with examples'. might you wish to consider shifting it further down to section 7 '6GCIoHE system for healthcare applications and challenges'?

Response: Done as per the suggestion ( use case is moved to section 7 -Application). See lines 650 to 655.

 6. Methodology section 4.0 revised and edited: Line # 359 and 392.

7. Sub-section heading 5.2.3 'network domain' is not formatted with Initial Caps (to align with the others).

Response: Corrected. Also formatting and titles are corrected throughout the article.

8. In sub-section 7.2 'challenges' you introduce six principles. you may wish to consider elaborating more on how you derive them. presently they do not sit very comfortably in the manuscript because (to my reading) nothing in the preceding sections builds explicitly to these six principles.

Response: Done. Section 7.1:  Applications: Lines 650 to 655 added.

Section 7.2 : Challenges: Lines 668 to 693.

 9. In sub-section 8.1 'theory's focal phenomena' you write "so far, the development of new theories was unnoticed within the discipline". first, i am not clear what exactly is meant by theories being unnoticed. second, even if i were to be clear, this comes across as a sweeping.

Response: Corrected (the line is removed).

10. In sub-section 8.2 'security, privacy...', you identify a gap, and elaborate on this gap in the entire first paragraph. i feel this is useful exposition and should be brought forward in your manuscript, say to sub-section 1.1 on 'problem formulation'. that is where your articulation of the gap would make (more) sense.

Response: Corrected. The paragraph is removed from 8.2 and moved to 1.1 sub-section. The gap is highlighted in 1.1 sub-section (see lines 93 to 98).

11. Still in sub-section 8.2, your third paragraph begins "the business model concepts underlying the disease". here again, it is not clear to me what you are trying to say. to my mind, i do not understand how a business model may underlie a disease.

Response: Corrected. Lines 730 to 738.

12, You end sub-section 8.2 with a colon ( : ), and you do so because the subsequent sub-sub-sections are a list. you may wish to consult with the Editor on the appropriate punctuation. i personally feel a full-stop / period ( . ) would be acceptable and appropriate.

Response: Done. Line # 738.

 13. Additional Revision:

Sub-section 1.3 is replaced by Table 1 with organization summary.

 14. Appendix A is added to describe an acronym list.

 15. The entire manuscript is edited and revised appropriately, and some figures are repositioned appropriately.

Reviewer 2 Report

Although the manuscript seems interesting in the first place. After reading it, I found it hard to define the real contributions of it.

First of all, the whole manuscript is excessively verbose and redundant. Just to start with, the very first sections with the context and related work are not well organized and hard to follow. Some type of summary diagram and better organization would be highly appreciated.
But the problem is that the rest of sections keep repeating over and over the same (or very similar concepts). It feels as if the whole article is about a survey, instead of a technical proposal. In the end the technical design and proposal of the authors is just one or two pages of the whole manuscript.
So a general recommendation would be: reorganize the text and keep it concise and relevant.

Related with the previous point. Section 3 is too chaotic in my opinion, there are subsections entitled CC links, others simply CC.
They seem to lack organization and they are really short to be followed. At some point, you get easily lost...
Also, Section 4 repeats text from the introduction (which should be removed), and Section 5 has strange formatting for the subsections (for example, 5.2.3 is on the left, and 5.2.4 has a space on the left, and no title?; but this is just an example of the many in the article).

One suggestion would be adding a table containing an acronym list to facilitate reading the article.

Additionally, most figures have very bad quality and/or are not really meaningful.
For example, Figure 8 is not really relevant for the article in my opinion. But, in general, all of them should be revised.

Finally, there are diverse formatting and English typos and errors. 
To mention a couple (but the whole text should be checked):
-Fig 7 portray... portrays! 
-and enable the collaboration... 

In conclusion, the article has potential, but it should be polished, with major changes, to be published.

Author Response

RESPONSE TO REVIEWER 2 REPORT

Appreciate your effort for reviewing the manuscript.

Proof reading including grammar/spelling was done using the following tools (a, b, c) and manually. The entire manuscript analysis ( originality, spelling, grammar, word choice, style, and vocabulary), was done using the tools (d, e, f, g) mentioned below:

  1. Powerful English Grammar Checker | SCRIBENS
  2. https://smallseotools.com/grammar-checker/
  3. https://cloud.trinka.ai/editor/main/02f66b9d-73a3-46ab-ae85-13f626ad72bd
  4. https://smallseotools.com/plagiarism-checker/
  5. https://www.check-plagiarism.com/article-rewriter
  6. https://www.paperrater.com/plagiarism_checker
  7. https://www.paperrater.com/free_paper_grader

Your comments are answered (in blue bold color) point by point below:

1. Although the manuscript seems interesting in the first place. After reading it, I found it hard to define the real contributions of it.

Response: The sub-section 1.2 is modified to highlight the significant contributions (see lines 100 to 109). The sub-section 8.7 also describes contribution of the theoretical framework. It should be noted that the development of theoretical framework for the 6GCIoHE system adoption and implementation is non-existent in the literature.

2. The whole manuscript is excessively verbose and redundant. Just to start with, the very first sections with the context and related work are not well organized and hard to follow. Some type of summary diagram and better organization would be highly appreciated.

Response: Done. Sub-section 1.3 is replaced by an organization summary Table 1.

3. But the problem is that the rest of sections keep repeating over and over the same (or very similar concepts). It feels as if the whole article is about a survey, instead of a technical proposal. In the end the technical design and proposal of the authors is just one or two pages of the whole manuscript.

Response: In this article, the focus is on the importance of theoretical framework to guide decision making around 6GCIoHE system technology, and mechanisms of technology interventions. Advances in the emerging 6G enabled Tactile Internet, CDI including tiny ML, and the “Internet of healthcare everything” (IoHE) is being poised to drive largely the next-gen healthcare evolution to create a phenomenon in the healthcare industry. A theoretical framework is an explanation of why a phenomenon occurs the way it does and reflects the body of new knowledge.

The significance of theory to guide the development, implementation, and evaluation of interventions has been supported by researchers in the healthcare fields. Yet the role of theory is not always clearly understood or fully recognized, especially in the healthcare research. Furthermore, theory-based interventions continue to be undervalued and underutilized in healthcare research. Hence, in our view, development of theoretical framework is a vital tool for guiding healthcare decision-making.

4. So a general recommendation would be: reorganize the text and keep it concise and relevant.

Response: Thanks for your recommendation.

The literature review and theoretical framework are intrinsically linked for logical understanding and developing the disparate, yet interconnected, essential parts of the literature review.

 Given both the diversity and growth of cutting-edge technologies in the healthcare ecosystem and healthcare researchers require frameworks to guide decision-making regarding their use.

 Each section and sub-sections describe distinctly the relevant concepts and frameworks pertaining to 6G, Tactile Internet, Cognitive Computing, MDA, Tiny ML and IoHE. Hence, relevant frameworks are conceptualized and described in the theoretical foundation (section 3).

5. Related with the previous point. Section 3 is too chaotic in my opinion, there are subsections entitled CC links, others simply CC.

Response: Revised to indicate Cognitive Computing (CC). With due respect, the frameworks and concepts are essential foundation for developing a novel theoretical framework.

Respectfully, in our view, this article is systematically organized and presented to achieve the primary purpose (to develop a theoretical framework for the 6GCIoHE system) of the research.

6. Section 4 repeats text from the introduction (which should be removed).

 Response:  Section 4 revised appropriately.  See the revised version: Lines: 359 to 392. Also,  section 4.1 related to PHM was removed.

7. Section 5 has strange formatting for the subsections (for example, 5.2.3 is on the left, and 5.2.4 has a space on the left, and no title; but this is just an example of the many in the article).

Response:  Formatting and titles are corrected throughout the article including sections 5.

8. One suggestion would be adding a table containing an acronym list to facilitate reading the article.

Response: Acronym list is added as suggested (see the section Appendix A)

 9. Additionally, most figures have very bad quality and/or are not meaningful.
For example, Figure 8 is not relevant for the article in my opinion. But, in general, all of them should be revised.

Response: May I point out that the figure 8 ( a figure is worth thousand words) provides the next leap evolution from mobile Internet, IoE, Tactile Internet  to 6G Tactile Internet which is meaningful and relevant to the research topic.

All figures are corrected to provide quality. Some figures are repositioned appropriately.

10. Finally, there are diverse formatting and English typos and errors. 
To mention a couple (but the whole text should be checked):
-Fig 7 portray... portrays! 
-and enable the collaboration.

Response: The entire text is proofread, and necessary corrections are made.

11. In conclusion, the article has potential, but it should be polished, with major changes, to be published.

Response: Appreciate your comments regarding the potential of the article. I have made the required and appropriate changes. Hopefully, my responses mentioned above satisfies your constructive comments.

12. Additional changes are made

Section 7.1:  Applications: Lines 650 to 655 added.

Section 7.2 : Challenges: Lines 668 to 693.

The entire manuscript is revised and edited appropriately.

Round 2

Reviewer 2 Report

The authors were really careful to address most of my comments successfully in my opinion. I really appreciate that they invested time in them and answered most of my concerns.

However, I'm still really concerned about the quality of the figures, which is really poor. I recommend that the authors use a different image editor in the future, there are many open source and free editors that will grant a better quality and improve a lot the overall presentation of the article.

This is not going to affect my decision, but I recommend it for future submissions if they don't have time at this point to modify the current one.

Author Response

I'm still really concerned about the quality of the figures, which is poor. I recommend that the authors use a different image editor in the future, there are many open source and free editors that will grant a better quality and improve a lot the overall presentation of the article.

Response:

In this revision, the quality of all the figures were addressed. Some of the figures are repositioned appropriately.

English spell check are done using the tools and by proofreading again.